# Multiple domains of scaffold Tudor protein play nonredundant roles in *Drosophila* germline

Samuel J Tindell[1], Alyssa G Boeving[1], Julia Aebersold[2], Alexey L Arkov[1]

Scaffold proteins play crucial roles in subcellular organization and function. In many organisms, proteins with multiple Tudor domains are required for the assembly of membraneless RNA–protein organelles (germ granules) in germ cells. Tudor domains are protein–protein interaction modules which bind to methylated polypeptides. *Drosophila* Tudor protein contains 11 Tudor domains, which is the highest number known in a single protein. The role of each of these domains in germ cell formation has not been systematically tested, and it is not clear if some domains are functionally redundant. Using CRISPR methodology, we generated mutations in several uncharacterized Tudor domains and showed that they all caused defects in germ cell formation. Mutations in individual domains affected Tudor protein differently, causing reduction in protein levels and defects in subcellular localization and in the assembly of germ granules. Our data suggest that multiple domains of Tudor protein are all needed for efficient germ cell formation, highlighting the rational for keeping many Tudor domains in protein scaffolds of biomolecular condensates in *Drosophila* and other organisms.

## Introduction

Scaffold proteins play major roles in different cells by bringing together their different partner proteins to initiate the effective response to cellular signals and induce and maintain formation of membraneless organelles (Good et al, 2011; Sanders et al, 2020; DiRusso et al, 2022). Scaffold proteins often contain several protein–protein interaction modules that allow them to bind to different partners during signal transduction or the assembly of biomolecular condensates. In particular, in germ cells of many animals, scaffold proteins containing multiple Tudor (Tud) domains play crucial roles in the assembly of RNA–protein membraneless organelles referred to as germ granules, which are involved in storage, spatial organization, and translational activation of mRNAs crucial for germline development (Arkov & Ramos, 2010; Voronina et al, 2011; Gao & Arkov, 2013; Westerich et al, 2023;

Chen et al, 2024; Pamula & Lehmann, 2024; Ramat et al, 2024). Canonical Tud domain is about 60–amino-acid $\beta$ barrel structure which has a binding pocket lined with aromatic amino acids that associate with methylated lysines or arginines of partner proteins (Simcikova et al, 2023). In some proteins, canonical Tud domain is inserted into an $\alpha$-helix/$\beta$-strand fold and these combined structural motifs are referred to as an extended Tud (eTud) domain (Shaw et al, 2007; Friberg et al, 2009; Liu et al, 2010).

*Drosophila* Tud protein contains the highest number of Tud domains known in a single protein (11) (Vo et al, 2019; Wahiduzzaman et al, 2024), and this polypeptide is absolutely required for germ granule assembly and primordial germ cell formation during early embryogenesis (Boswell & Mahowald, 1985; Thomson & Lasko, 2004; Arkov et al, 2006). In particular, in germ granules, Tud protein directly interacts with methylated Piwi protein Aubergine (Aub) and methylated glycolytic enzyme pyruvate kinase (PyK) (Liu et al, 2010; Gao et al, 2015; Vo et al, 2019; Wahiduzzaman et al, 2024). Previous work solved structures of Tud domains 9–11 in Tud protein, which revealed their eTud architecture, and suggested that other domains of Tud may form an eTud configuration; however, canonical Tud domain in eTud is a principal protein–protein interaction module (Liu et al, 2010; Ren et al, 2014).

It has not been determined whether all 11 Tud domains of Tud protein play a role in its structure and function during germ cell development or some domains are redundant or exclusively used in non-germline (somatic) cell types (Tindell et al, 2020).

Using forward genetics maternal mutant screens and directed mutagenesis of the *tud* transgene, which expressed C-terminal fragment of Tud, previous studies characterized mutations in several Tud domains (Boswell & Mahowald, 1985; Arkov et al, 2006; Liu et al, 2010). Virtually, all these mutations caused defects in germ cell formation and decreased the levels of Tud protein or changed the morphology, decreased the size, and the number of germ granules (polar granules) assembled in posterior cytoplasm of the egg (germ plasm). However, there was a set of the N-terminal Tud domains (domains 2–6), which have not been characterized with previous mutational approaches, and it has not been clear whether these domains contribute to the structure and function of Tud scaffold in germ cell formation.

[1]Department of Biological Sciences, Murray State University, Murray, KY, USA   [2]Micro/Nano Technology Center, University of Louisville, Louisville, KY, USA

Correspondence: aarkov@murraystate.edu

In this work, we asked whether, similarly to other previously studied Tud domains of Tud scaffold, most of these poorly characterized N-terminal domains individually contribute to the structure and function of Tud protein in a single developmental process of primordial germ cell formation. To this end, using CRISPR/Cas9 methodology, we introduced deletions of each of the canonical Tud domains 2–5 sequences in the native *tud* locus and tested whether these deletions cause defects in germ cell formation, Tud expression levels, Tud localization to the posterior pole of the early embryos, and polar granule assembly.

Our data showed that each of these poorly characterized single domains contributes to germ cell formation by regulating Tud amounts in the germline, Tud localization to the germ plasm, and the size of polar granules. Overall, this work and previous research provide evidence for the importance of nearly all Tud domains of Tud scaffold for the protein structure and function and suggests that each of the multiple Tud domains of Tud protein is used during germline development in *Drosophila*. Surprisingly, there appears to be no redundancy of different Tud domains in Tud scaffold during its involvement in the formation of primordial germ cells in early *Drosophila* embryo.

# Results

### Tud domains 2–5 mutants show reduction in the number of primordial germ cells in *Drosophila* embryos

Previous work has characterized mutations in several canonical Tud domain sequences of Tud protein and helped establish the importance of specific individual domains of this scaffold protein for primordial germ cell formation, binding to polar granule components, polar granule assembly, and morphology. In particular, mutations in Tud domains 1 (Arkov et al, 2006) and 7–11 (Boswell & Mahowald, 1985; Arkov et al, 2006; Liu et al, 2010) (Fig S1A and B) caused reduction in the number of primordial germ cells formed in posterior of early embryos or germ plasm localization of Tud domain–binding protein Aub. In addition, the germ plasm of Tud domains 1 (*tud*^A36^), 7 (*tud*^4^), and 10 (*tud*^B42^) mutants (Fig S1A and B) was examined with electron microscopy (EM), and the reduction in size, number, or abnormal morphology of the polar granules were detected in these mutants (Boswell & Mahowald, 1985; Arkov et al, 2006).

To obtain the comprehensive understanding of how Tud protein uses its Tud domains during polar granule assembly and germ cell formation, using CRISPR methodology, we generated small deletions removing most of the remaining canonical domains that have not been characterized previously (domains 2–5) in native *tud* locus (Fig 1A and B). Although, similarly to Tud domains 2–5, Tud domain 6 was not well characterized, we could not generate a similar deletion of this domain despite rigorous efforts.

First, we determined if each of the mutants shows reduction in the number of primordial germ cells formed in the embryos. Fig 2A–F shows that each single Tud domain deletion caused significant reduction in the number of germ cells. The strongest germ cell formation phenotypes were exhibited by Tud domain 3 mutant

(*tud*^dom3^; no germ cells formed in 100% embryos examined) (Fig 2C and F) and Tud domain 4 mutant (*tud*^dom4^; only 12% embryos formed some germ cells) (Fig 2D and F). Tud domain 2 (*tud*^dom2^) and 5 (*tud*^dom5^) mutants showed less strong but still significant reduction in the number of primordial germ cells (about twofold reduction in germ cell number compared with the WT control) (Fig 2A, B, E, and F). These data indicate that presence of each domain is needed to maximize the involvement of Tud scaffold in germ cell formation.

### Mutations in Tudor domains affect Tudor protein enrichment in the germ plasm and protein expression levels

Defects in germ cell formation might be caused by the inability of Tud protein to reach the germ plasm, where it is normally strongly localized (Bardsley et al, 1993; Arkov et al, 2006; Zheng et al, 2016), or its reduced stability. Therefore, we determined whether Tud domain mutant proteins can be detected in the germ plasm of early preblastoderm embryos before germ cell formation. Fig 3 shows that Tud domains 2, 3, and 5 mutants fail to enrich Tud in germ plasm labeled with anti-Vasa (Vas) antibody (Fig 3B, C, and E). Contrary to this, Tud domain 4 mutant protein was able to specifically localize to the germ plasm (Fig 3D) similar to the WT control (Fig 3A).

Next, we asked if *tud* mutants affected Tud protein levels. To this end, to determine whether the mutations decreased Tud stability or expression levels, we used our new anti-Tud antibody (Fig S2A and B) and anti-FLAG antibody because all CRISPR-edited *tud* alleles encoded N-terminal FLAG tag (Fig 4A–C). Although Tud protein amounts were reduced to about 40–60% of the WT control Tud levels in Tud domains 2, 4, and 5 mutants, this reduction cannot explain the lack of Tud accumulation in the germ plasm of Tud domains 2 and 5 proteins (Fig 3B and E) because similarly expressed Tud domain 4 mutant protein (49% expression of the WT control) shows robust Tud enrichment in the germ plasm (Fig 3D). Therefore, Tud domains 2 and 5 may contribute to the localization or maintenance of Tud in the posterior germ plasm.

Tud domain 3 mutant protein failed to be expressed at detectable levels (Fig 4A–C), indicating that the mutation caused dramatic decrease in protein stability consistent with very strong germ cell formation phenotype for this mutant (Fig 2C and F) and apparent lack of the mutant protein in the germ plasm (Fig 3C). This is a striking result given that in this Tud domain mutant, only 1 out of 11 Tud domains is deleted, and it resembles previous data demonstrating a strong reduction in Tud protein amount when two aromatic amino acids in a single Tud domain 7 were mutated in the Tud fragment containing Tud domains 7–11 (Fig S1A and B) (Liu et al, 2010).

### Tud domain mutants affect polar granule assembly

The expression of Tud protein is essential for the assembly of polar granules, composed of RNA and proteins required for germ cell formation (Boswell & Mahowald, 1985; Thomson & Lasko, 2004; Arkov et al, 2006). Therefore, using super-resolution microscopy imaging (Airyscan) (Bond et al, 2022) and EM, we tested if Tud domains 2, 4, and 5 mutants, which all express Tud protein (Fig 4)

## A

### Tudor protein and its 11 Tudor domains

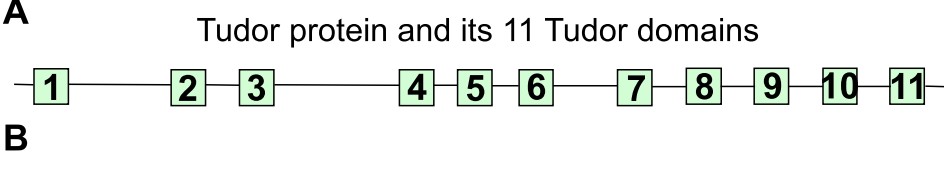

## B

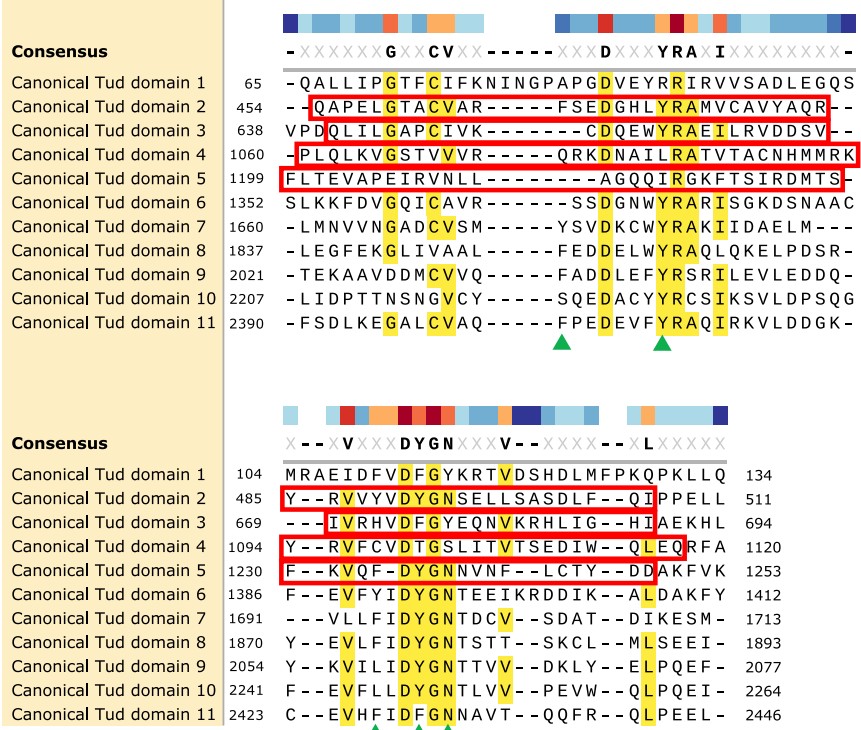

**Figure 1. Tudor domains of Tudor scaffold protein and indicated deletion mutants in domains 2–5 generated by CRISPR methodology.**

**(A)** A diagram of Tud protein with its 11 Tud domains (squares) at their approximate locations is indicated. **(B)** Alignment of canonical Tud domain sequences of Tud protein. Amino acid boundaries of each domain are specified in the alignment. Conserved or similar residues are indicated in yellow and in consensus sequence. Poorly characterized Tud domains 2–5, analyzed in this study, were deleted by CRISPR methodology in the native *tud* locus and these deletions are outlined with red boxes. Green triangles point to the residues of the binding pocket of canonical Tud domain 11 structure which binds to methylated arginine of Piwi protein Aubergine (Aub) (Liu et al, 2010). Different Tud domains show significant variations including those in the indicated positions of the aromatic amino acids in the binding pocket of Tud domain 11 ("aromatic cage"). Although some amino acid changes in the binding pocket allow binding of Aub as shown for Tud domains 1, 3, 4, 6, and 9, variations in the binding pocket of other domains may prevent their interactions with methylated Aub as observed for Tud domain 10 suggesting binding of unmethylated partner proteins to some Tud domains (Ren et al, 2014; Vo et al, 2019).

but are defective in germ cell formation (Fig 2), show defects in polar granules size or morphology.

First, using super-resolution microscopy imaging, we detected incorporation of Tud and Vas proteins in small granules at the posterior poles of early embryos in all these Tud domain mutants (Fig 5A–D, three left panels). Quantification of colocalization of Vas and Tud indicated that mutants appear to have less Vas granules that colocalize with Tud compared with WT control; however, this difference is not statistically significant (Table S1).

Although the super-resolution microscopy imaging indicated the assembly of mutant Tud proteins into the granules, it can be difficult to characterize their size and morphology accurately if they are too small and, therefore, diffraction-limited for our Airyscan super-resolution modality which has the lateral spatial resolution limit of about 140 nm. Therefore, to overcome these limitations, we used the EM approach. In WT germ plasm, polar granules were easily detected as characteristic amorphous membraneless electron-dense particles (Fig 5A, right panel). Furthermore, Tud domains 2, 4, and 5 mutants clearly assemble electron-dense polar granules, which, however, were significantly smaller than in the WT control (Fig 5B–E). Therefore, despite the apparent ability of Tud to be incorporated in the granules in the germ plasm in all these Tud domain mutants, the mutations cause 3.4–4.5-fold reduction in size

(area) of electron-dense polar granules compared with the granules of WT control embryos (Fig 5E).

The assembly of multiple small electron-dense granules in Tud domains 2 and 5 mutants, which show defective Tud localization to the germ plasm (Fig 3B and E), is consistent with our super-resolution microscopy detection of Tud protein in the germ plasm in these mutants because germ plasm Tud is required for the formation of polar granules. Therefore, our data presented in Figs 3 and 5, which include fluorescent microscopy and EM data, suggest that although Tud is not enriched in the germ plasm in Tud domains 2 and 5 mutants, it is still present there at the levels sufficient to drive the assembly of abnormal germ granules.

## Discussion

In this work, we provide evidence for nonredundant roles of multiple domains of Tud scaffold proteins in primordial germ cell formation in *Drosophila*. Tud protein contains 11 domains and, based on previous research and this work, nearly all Tud domains of this protein are now shown to contribute to this specific developmental process during germline development (Table 1). Therefore, in regard to Tud domains of Tud protein, the term

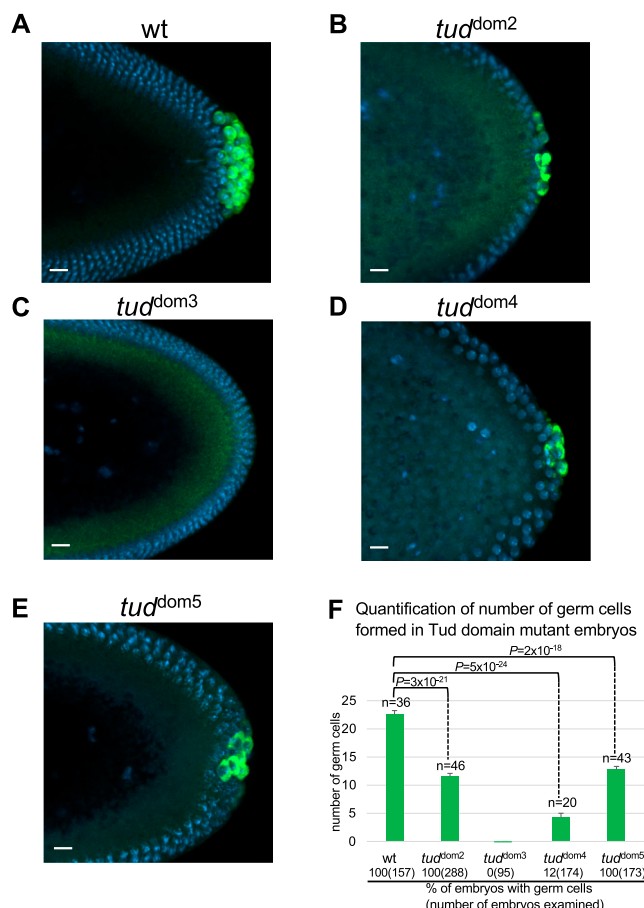

**Figure 2. Mutations in Tudor domains 2–5 of Tudor scaffold protein cause defects in primordial germ cell formation in *Drosophila* embryos.**
**(A, B, C, D, E)** Germ cells at posterior pole of early (stages 4–5) embryos are labeled with anti-Vasa (Vas) antibody (green). DAPI labels nuclei (blue). **(A)** *wt* control embryos were produced by females with *wt tud* allele/*tud* deletion (*Df(2R)Pu$^{rP133}$*, see the Materials and Methods section). This *wt tud* allele was tagged with FLAG tag in the native *tud* locus with CRISPR editing (Tindell et al, 2020) and was used for the production of all Tud domain mutants with CRISPR methodology reported in this work. **(B, C, D, E)** Mutant embryos were generated by females transheterozygous for the indicated Tud domain mutation and *tud* deletion *Df(2R)Pu$^{rP133}$*. **(A, B, C, D, E, F)** Quantification of germ cells' number (mean ± s.e.m) formed in different Tud domain mutant embryos and *wt* control whose representative images are shown in (A, B, C, D, E). Germ cells were counted from multiple embryos (*n*) at stages 10–13 of embryonic development. During these stages, the germ cells are spread out to a maximal degree inside the embryo and can be counted most accurately (Santos & Lehmann, 2004). Germ cell numbers for *wt*, *tud$^{dom2}$*, *tud$^{dom4}$*, and *tud$^{dom5}$* mutants were 22.6 ± 0.7, 11.5 ± 0.6, 4.3 ± 0.7, and 12.8 ± 0.5, respectively. Reduction of germ cells number in all the mutants compared with *wt* control was statistically significant (unpaired two-tailed *t* test was used; *P*-values indicated). Separately, for each mutant, the percentage of embryos at stages 5–14 that contain any germ cells was scored and indicated at the bottom of the figure. Although all *wt*, *tud$^{dom2}$*, and *tud$^{dom5}$* embryos showed germ cells, only 12% of *tud$^{dom4}$* embryos formed germ cells and *tud$^{dom3}$* embryos failed to form any germ cells. Number of embryos scored is indicated in parentheses following the percentage value. In (A, B, C, D, E) scale bars are 10 *μ*m.

"nonredundancy" is used here to indicate that any given single Tud domain contributes to the normal function and expression of Tud protein because mutations in any of these domains result in significant defects during germ cell development.

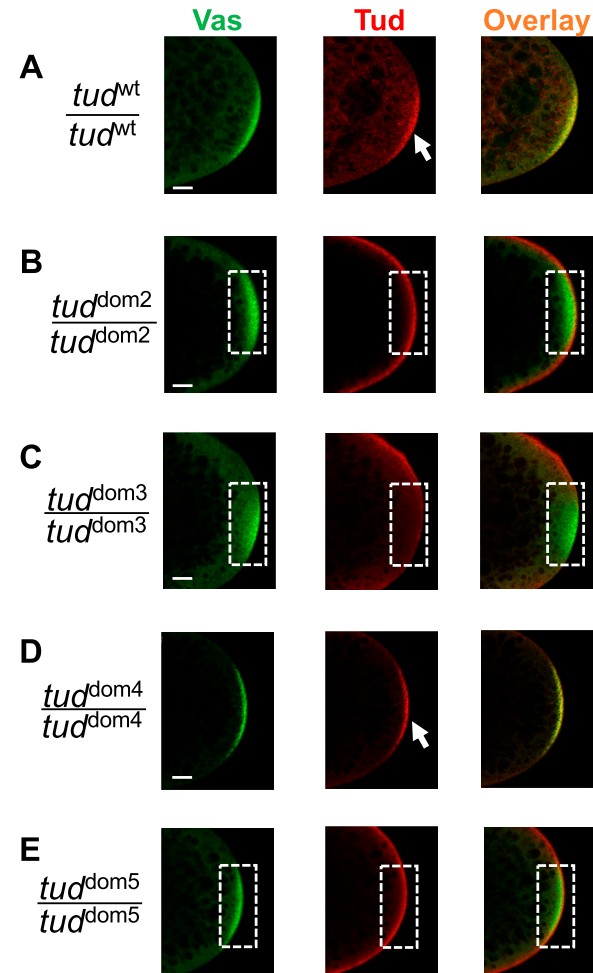

**Figure 3. Different Tudor domain mutants show different levels of enrichment of Tudor protein in germ plasm of early embryos.**
**(A, B, C, D, E)** Representative confocal microscopy optical sections of posterior poles of early (stages 1–2) embryos from homozygous *wt flag-tud* (A) or Tud domain mutant (B, C, D, E) females. The embryos were immunostained with antibodies against Vas protein (green) to label germ plasm and anti-FLAG antibody to label Tud (red). Overlay images are shown. Whereas *wt* and Tud domain 4 mutant embryos show the localization of Tud protein to the germ plasm (arrows), Tud domains 2, 3, and 5 mutants fail to show Tud enrichment there (germ plasm area is outlined with dashed line), suggesting defects in localization to/maintenance in the germ plasm or in the expression of Tud protein. For each mutant and WT control experiments, z-stacks (18–49 optical sections in each stack) for each of the 9–10 embryos per genotype were acquired for analysis of protein distribution as follows. All *wt* (n = 9) and all Tud domain 4 mutant (n = 9) embryos showed Tud enrichment in the germ plasm, only two Tud domain 2 embryos (n = 10) showed enrichment, and no Tud domain 3 mutant (n = 9) and no Tud domain 5 mutant (n = 9) embryos showed Tud enrichment. Scale bars are 15 *μ*m.

In general, during formation of biomolecular condensates, multiple protein–protein interaction modules can play an important role in providing sufficient valence for multiple interactions with components of the condensates that drive their assembly (Sanders et al, 2020). Although Tud scaffold may require a certain number of Tud domains to ensure sufficient valence for germ granule assembly, here we report strong phenotypic effects of mutations in single domains of Tud that are not likely to be

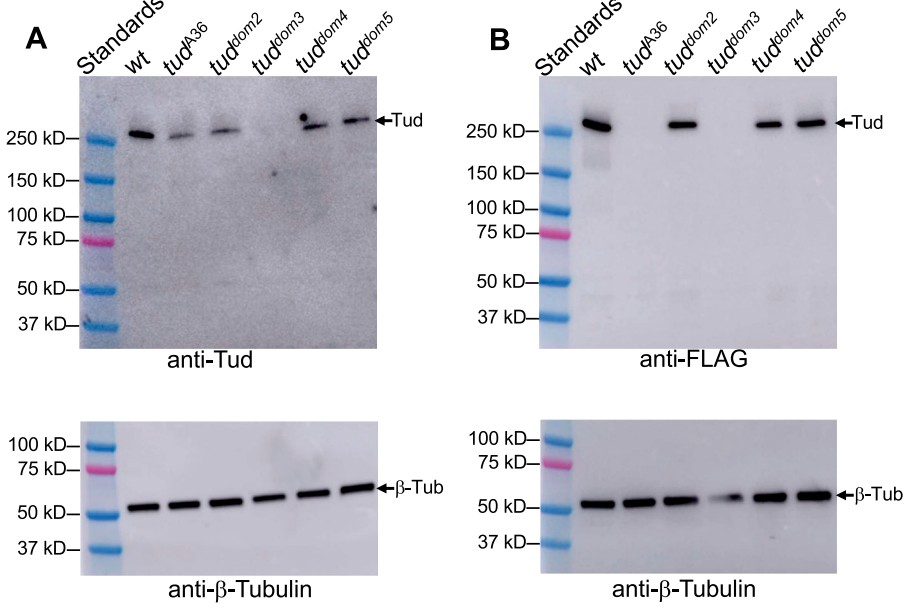

**Figure 4. Expression levels of Tud domain mutants.**
**(A, B)** Western blot data show expression levels of Tud protein in *wt* control and different Tud domain mutant ovaries isolated from females transheterozygous for a *wt* control allele (*flag-tud*), or given *tud* mutant allele, and *tud* deletion *Df(2R) Pu^rP133*. In addition to Tud domains 2–5 mutant and the *wt tud* control genes, which all encode the N-terminal FLAG tag, previously characterized and quantified Tud domain 1 mutant (*tud^A36*) (Arkov et al, 2006) (Fig S1, Table 1) was included in the Western blot experiments. This *tud^A36* does not have a FLAG tag. **(A)** Top panel: Western blot experiment with anti-Tud antibody (validated in Fig S2) detects Tud expression in all mutants except for Tud domain 3 mutant. Bottom panel: Sample loading was controlled with anti–β-Tubulin antibody using the same gel shown in the top panel. **(B)** A different Western blot experiment with anti-FLAG antibody supports data obtained with anti-Tud antibody. Tagless *tud^A36* mutant protein serves as a negative control in this experiment. Top and bottom panels show Western blot data with anti-FLAG and anti–β-Tubulin antibody (loading control), respectively. **(C)** Expression levels of mutant Tud domain proteins were quantified as percentages of *tud* expression in *wt* controls (mean ± s.e.m) from four biological replicate Western blot experiments using anti-Tud antibody.

### C

Quantification of Tudor protein levels in Tud domain mutants

| Tud domain mutant | % of wt control [mean±s.e.m (n)] |
|---|---|
| *tud^dom2* | 39.2±4.1 (4) |
| *tud^dom3* | Not detected (4) |
| *tud^dom4* | 49.2±12.7 (4) |
| *tud^dom5* | 61.8±9.4 (4) |

caused merely by the reduction of Tud valence because in each mutant, there are still 10 Tud domains remaining. In addition, we provide evidence that mutations in certain Tud domains have distinct phenotypes, indicating specific roles of these domains in germ cell formation.

Previous work showed the functional importance of Tud domains 1, 7–11 in different aspects of germ cell formation, interactions with methylated Tud protein–binding partners Aub and pyruvate kinase (PyK) and the functional assembly of polar granules (Arkov et al, 2006; Liu et al, 2010; Gao et al, 2015; Vo et al, 2019; Ramat et al, 2024; Wahiduzzaman et al, 2024). However, apart from the contribution of some of the N-terminal Tud domains 2–6 in Aub binding detected in vitro (Creed et al, 2010; Vo et al, 2019), the in vivo role of these domains in germ cell formation remained unknown, and it was not clear whether there is a functional redundancy among Tud domains of Tud scaffold in germ cell formation given their remarkably high number in a single protein. Also, because Tud protein was detected in soma (brain glia) (Tindell et al, 2020), one could propose that there are different subsets of Tud domains, each specific for Tud function in either soma or germline.

In this work, we generated small deletions in *tud* locus with CRISPR methodology that separately removed Tud domains 2–5, and characterized these novel mutants in detail using phenotypic analysis, super-resolution microscopy, and EM. All these mutations caused defects in primordial germ cell formation.

An important consideration for defects in germ cell formation that we detected in Tud domain mutants is the possibility that these defects are caused by a reduction in Tud protein levels in mutants. Although reduction in Tud protein amounts is observed in the mutants and it could contribute to the mutant phenotype, we suggest that this protein reduction is not the only cause for the observed functional defects in germ cell formation caused by many Tud domain mutations due to the following reasons. First, reduction of *tud* dosage to one copy of the gene causes virtually no reduction in the number of primordial germ cells formed in early fly embryos (Thomson et al, 2008), indicating that germ cell formation can proceed efficiently with Tud generated from just one copy of *tud* gene. Second, we demonstrated that despite similar reduction in Tud protein levels, different Tud domain mutants had different specific functional effects arguing against a common cause of reduced protein expression for defective germ cell formation observed in all these mutants. Specifically, Tud domain 2 and 5 mutant embryos

Polar granules in the germ plasm at posterior pole of early *Drosophila* embryos

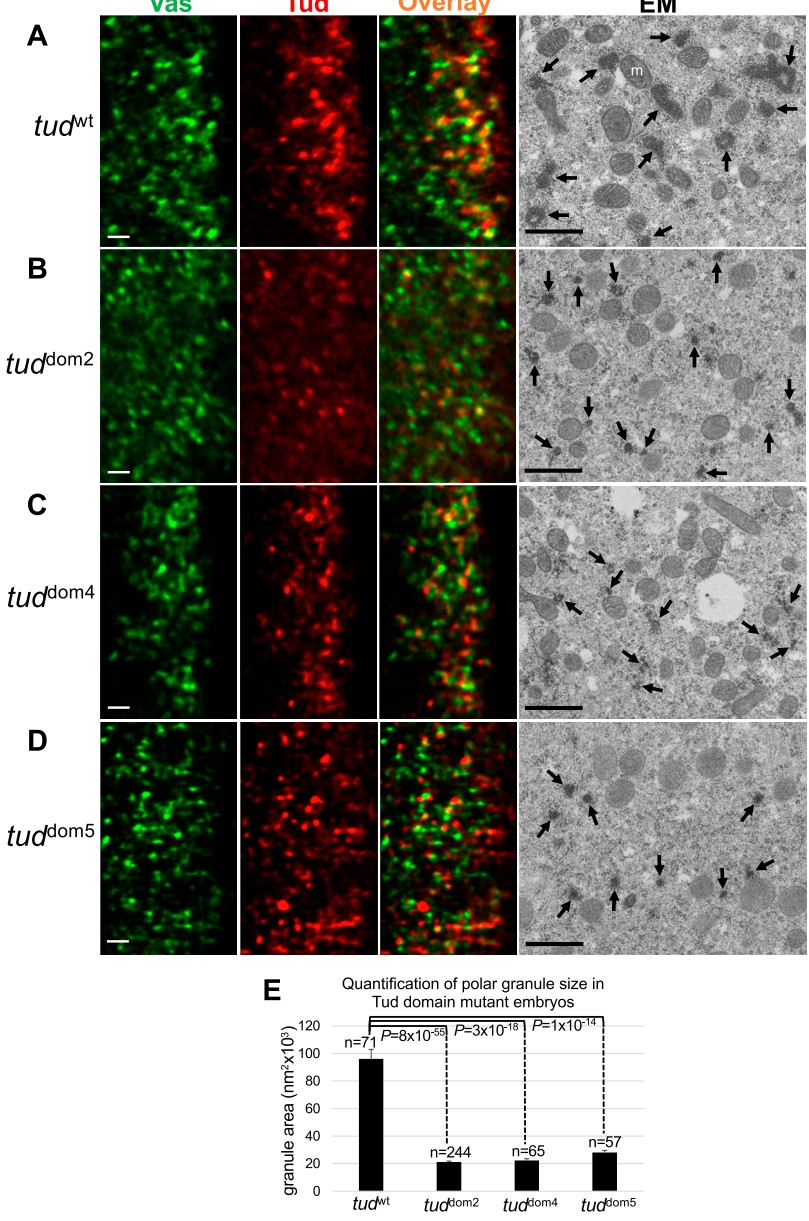

A  *tud*wt

B  *tud*dom2

C  *tud*dom4

D  *tud*dom5

E  Quantification of polar granule size in Tud domain mutant embryos

**Figure 5. Tudor domain mutants assemble small electron-dense polar granules in germ plasm at posterior pole of early *Drosophila* embryos.**
**(A, B, C, D)** Germ plasm at the posterior pole of early embryos (stages 1–2) was imaged with super-resolution microscopy (left three columns) and transmission electron microscopy (EM, right column) for *wt* control (A) and indicated Tud domains 2, 4, and 5 mutants (B, C, D) shown to express Tud protein (Fig 4). In these experiments, *wt* control, or mutant embryos, were from females transheterozygous for *wt* control allele (*flag-tud*), or indicated *tud* mutant allele, and *tud* deletion *Df(2R)Pu*rP133 and all Tud proteins had FLAG tag. For super-resolution experiments, embryos were immunostained with anti-Vas (green) and anti-FLAG (red) antibodies. Both Vas and Tud are assembled into granules in the germ plasm of the mutant embryos. Representative super-resolution optical sections for each mutant and *wt* control are shown. For super-resolution imaging, z-stacks for 8–10 embryos per genotype (21–101 optical sections per embryo) were analyzed. EM data further revealed electron-dense polar granules assembled in the mutants that were smaller than those in the *wt* control (indicated with arrows). **(E)** Quantification of polar granule size (average area ± s.e.m) from electron micrographs for the Tud domain mutants and *wt* embryos shows significant decrease in size of the granules in each of the mutants compared with the *wt* (unpaired two-tailed *t* test was used; *P*-values and numbers of granules (n) measured for each genotype are indicated). Average granule size (nm$^2$ × 10$^3$) in *wt*, Tud domain 2, Tud domain 4, and Tud domain 5 mutants is 96.1 ± 6.9, 21.2 ± 0.7, 22.3 ± 1.3, and 28.0 ± 1.6, respectively. Scale bars for all images are 1 *μ*m, m, mitochondria.

showed defects in Tud enrichment in the posterior pole. In contrast to Tud domains 2 and 5 mutants, Tud domain 4 mutant showed strong enrichment in the germ plasm. However, despite this germ plasm enrichment and contrary to Tud domains 2 and 5 mutants, Tud domain 4 mutation caused a very strong reduction in the number of germ cells. The difference in the strength of the germ cell formation phenotype among Tud domain 4 mutant and the other mutants cannot be explained by the different expression levels of Tud since Tud domain 2-, 4-, and 5-mutant proteins are expressed at similar amounts (Fig 4C). Our data suggest distinct functions of the N-terminal Tud domains: Tud domains 2 and 5 might be similarly involved in Tud protein

localization to the germ plasm and Tud domain 4 is needed downstream of Tud localization step leading to germ cell formation.

Interestingly, deletion of Tud domain 3 resulted in no detectable Tud protein expression, which explains complete absence of germ cells in this mutant. This is a surprising result, which suggests a critical function of this single 57–amino-acid Tud domain in expression and stability of 2,515–amino-acid Tud protein, which could be defective in this mutant because of several reasons. These reasons may include inability of this mutant Tud to associate with the Tud domain 3–specific interacting partner protein needed to stabilize Tud, the crucial function of the domain in the assembly

**Table 1. Role of Tudor domains in Tudor scaffold protein.**

| Tud domain | Mutant | Effect of mutation | Binding to other proteins | References |
|---|---|---|---|---|
| 1 | Arg91Trp (tud[A36]) | Reduction in number of germ cells[a]; abnormal morphology of polar granules; small polar granules[b] | Aubergine | Arkov et al (2006), Vo et al (2019), and Ramat et al (2024) |
| 2 | △(Gln454-Ile506) (tud[dom2]) | Reduction in number of germ cells; small polar granules | Tandem Tud domains 1 and 2 bind Pyruvate Kinase | This study; (Wahiduzzaman et al, 2024) |
| 3 | △(Gln641-Ile689) (tud[dom3]) | Reduction in number of germ cells; no Tud protein detected | Aubergine | This study; (Vo et al, 2019) |
| 4 | △(Pro1060-Gln1117) (tud[dom4]) | Reduction in number of germ cells; small polar granules | Aubergine | This study; (Vo et al, 2019) |
| 5 | △(Phe1199-Asp1248) (tud[dom5]) | Reduction in number of germ cells; small polar granules | No interacting proteins reported | This study |
| 6 | No data reported | | Aubergine | Vo et al (2019) |
| 7 | Tyr1680Ala, Tyr1697Ala (tud[D7*]) | Reduction in number of germ cells; strong decrease in Tud protein levels | Aubergine | Liu et al (2010) and Ren et al (2014) |
| 7 | △(Asp1708-Lys1710) (tud[4]) | Defective germ cells; small polar granules | Aubergine | Boswell & Mahowald (1985) and Arkov et al (2006) |
| 8 | Tyr1857Ala, Tyr1877Ala (tud[D8*]) | Reduced localization of Aubergine in germ plasm | Aubergine; tandem Tud domains 7 and 8 bind Pyruvate Kinase | Liu et al (2010), Ren et al (2014), and Wahiduzzaman et al (2024) |
| 9 | Tyr2041Ala Tyr2061Ala (tud[D9*]) | Reduction in number of germ cells | Aubergine; tandem Tud domains 8 and 9 bind Pyruvate Kinase | Liu et al (2010), Ren et al (2014), Vo et al (2019), and Wahiduzzaman et al (2024) |
| 10 | Tyr2227Ala Tyr2248Ala (tud[D10*]) | Reduction in number of germ cells | No interacting proteins reported with the single domain | Liu et al (2010) |
| 10 | Arg2228Cys (tud[B42]) | Reduction in number of germ cells; small polar granules | No interacting proteins reported with the single domain | Arkov et al (2006) |
| 11 | Tyr2410Ala Phe2430Ala (tud[D11*]) | Reduction in number of germ cells | Aubergine; tandem Tud domains 10 and 11 bind Pyruvate Kinase | Liu et al (2010), Vo et al (2019), and Wahiduzzaman et al (2024) |

[a]Reduction in the number of primordial germ cells formed in early *Drosophila* embryos generated by *tud* mutant mothers is indicated for a given mutant regardless of the strength of the observed maternal germ cell formation phenotype.
[b]If indicated, abnormal morphology or decrease in size of polar granules was observed using EM.

of Tud in germ granules or in demixing of Tud molecules from the cytosol into the granules.

In all *tud* mutants, which were examined for the presence of polar granules, these membraneless organelles are either extremely difficult to find or smaller or of abnormal morphology than in the WT control (Boswell & Mahowald, 1985; Thomson & Lasko, 2004; Arkov et al, 2006). In this work, using the EM approach to visualize polar granules, we found that all Tud protein–expressing Tud domains 2, 4, and 5 mutants formed small electron-dense polar granules. Furthermore, Tud-containing granules are detected in the germ plasm of these mutant embryos with super-resolution microscopy imaging (Fig 5), suggesting that mutant Tud proteins can be assembled into polar granules. The size reduction of polar granules in the mutants may contribute to the reduction in germ cell number detected in these mutants as the granule components are essential for germline development. At the same time, small polar granules formed in the Tud domain mutants may be functionally deficient in translational regulation of germline RNAs leading to defects in germ cell development. Consistent with this idea, abnormal and small polar granules formed in previously described Tud domain 1 mutant, *tud*[A36] (Arkov et al, 2006) (Table 1), were defective in translational activation of germline *nanos* RNA (Ramat et al, 2024). Because single

Tud domains and their methylated ligands can drive the formation of biomolecular condensates (Courchaine et al, 2021), it is conceivable that the lack of certain single Tud domains in Tud scaffold may hinder the formation of polar granules. However, it is remarkable that a single domain has a noticeable effect on polar granule assembly given that 10 remaining Tud domains are still present in the Tud scaffold.

Future research will provide biochemical and structural understanding of how all Tud domains and their specific binding partners work together on Tud scaffold as one functional system and shed light on molecular rational for the nonredundancy of multiple Tud domains indicated by this work.

# Materials and Methods

### Tudor domain mutants

Deletions of Tud domains 2–5 were generated in the native *tud* locus that was previously tagged with N-terminal FLAG tag (Tindell et al, 2020) using CRISPR/Cas9 methodology. The Tud domain deletions and the corresponding *tud* alleles are shown in Fig S1

and are as follows: $tud^{dom2}$: Gln454-Ile506; $tud^{dom3}$: Gln641-Ile689; $tud^{dom4}$: Pro1060-Gln1117; and $tud^{dom5}$: Phe1199-Asp1248. All deletions were generated using oligonucleotide (ODN) donor templates for CRISPR/Cas9–mediated homology-directed repair (HDR) as previously described (Gratz et al, 2015). In particular, for generation of Tud domain deletions, constructs for two guide RNAs (gRNAs) targeting *tud* coding region at the beginning and end of each Tud domain were injected into early *Drosophila* embryos containing Cas9 maternally expressed from *nanos* (*nos*) promoter (from Bloomington *Drosophila* Stock Center, BDSC stock #54591) (*tud* locus sequences targeted by all gRNAs for each Tud domain are provided in Table S2). Each of the targeting sites had a protospacer adjacent motif (PAM) required for Cas9-induced double-strand break (Table S2). Balanced mutant lines were generated by Rainbow Transgenic Flies, Inc. and confirmed with sequencing. Unless specified otherwise, for most experiments described in this work, embryos or ovaries were from females transheterozygous for a *tud* allele and a *tud* deletion, $Df(2R)Pu^{rP133}$.

## Production of rabbit anti-Tud antibody

Rabbit anti-Tud antibody was raised against purified C-terminal Tud protein fragment containing Tud domains 7–11 (amino acids 1,605–2,515). This Tud fragment was produced from pET SUMO vector in *E. coli* BL21 (DE3) cells as described previously (Wahiduzzaman et al, 2024). The antibody was produced by Cocalico Biologicals according to the standard protocol detailed previously (Kharel et al, 2024). Subsequently, the antibody was validated and confirmed to recognize Tud protein specifically in both Western blot (Fig 4) and immunohistochemistry (Fig S2) experiments. For immunohistochemistry, 1:1,500 dilution of the antibody was used.

## Immunohistochemistry

These methods have been described by Stein et al (2002) and Navarro et al (2004). For whole-mount immunostaining of fly embryos, rabbit anti-Vasa antibody (1:1,000) (Stein et al, 2002; Trcek et al, 2015; Zheng et al, 2016) was used. Also, to detect Tud, mouse anti-FLAG antibody (1:2,500; Millipore Sigma) (Wahiduzzaman et al, 2024) was used.

For quantification of primordial germ cells in Tud domain mutants and WT control embryos, germ cells were immunolabeled with the anti-Vasa antibody and manually counted at stages 10–13 of embryonic development (Thomson et al, 2008). Unpaired two-tailed *t* test was used to evaluate statistical significance of difference in number of germ cells formed in mutants compared with WT control (Fig 2F).

## Analysis of mutant Tud protein expression

Expression levels of Tud protein in ovarian extracts from *tud* mutants and WT control were quantified with the Western blot procedure as detailed previously (Arkov et al, 2006). The following antibodies were used for detection of Tud protein: rabbit anti-Tud validated in this work (1:1,500), mouse anti-FLAG antibody (1:3,000;

Millipore Sigma) as alternative antibody to confirm Tud expression, and mouse anti-β-Tubulin antibody (1:5,000; Millipore Sigma) as a loading control.

## Super-resolution microscopy

Super-resolution microscopy imaging was carried out essentially as described (Wahiduzzaman et al, 2024). In particular, a Zeiss LSM 980/Airyscan super-resolution module system, inverted laser scanning confocal microscope AxioObserver, and Plan-Apochromat x63/1.4 Oil DIC M27 objective were used. For every experiment, the images were acquired equally for mutants and WT control as z-stacks and subsequently analyzed with Imaris software (version 9.5, Oxford Instruments) and an HP Z8 workstation. Methodology used for Vas/Tud granule colocalization analysis, which determined the co-occurrence of both proteins in the same polar granules using Imaris software (Table S1), was described previously (Kharel et al, 2024). Unpaired two-tailed *t* test was used for evaluation of statistical significance of differences between Vas/Tud colocalization values for Tud domain mutants and WT control.

## Electron microscopy and quantification of polar granule size

Preparation of *Drosophila* embryos to image the posterior germ plasm and polar granules with EM was carried out essentially as described (Arkov et al, 2006). In particular, after initial fixation of the dechorionated embryos in heptane saturated with 12.5% glutaraldehyde for 20 min at room temperature, vitelline membranes were removed manually. Then, the embryos were fixed in 2% paraformaldehyde/2.5% glutaraldehyde in 0.1 M sodium cacodylate buffer (pH 7.4) overnight at 5°C. The fixed embryos were rinsed three times for 10 min with 0.1 M sodium cacodylate buffer followed by staining with 1% osmium tetroxide for 1 h. The samples were rinsed one time for 10 min in 0.1 M sodium cacodylate buffer and stained en bloc with 1% uranyl acetate for 30 min. The samples were rinsed twice with distilled water and then dehydrated progressively with 30%, 50%, 70%, 90%, and 95% ethanol for 10 min and three times with anhydrous ethanol for 15 min. Embedding was performed with 1:1 Spurr's resin and anhydrous ethanol along with an increased concentration of 3:1 resin and anhydrous ethanol for 1 h. The samples were placed in 100% resin under vacuum for 24 h without agitation and then cured at 70°C for 24 h. The embedded samples were sectioned to 80 nm with a Leica UC7 ultramicrotome and placed onto nickel slot grids with formvar support film. Post lead citrate staining was performed for 2 min on the grids and followed by rinsing with DI water. The samples were imaged with a Hitachi HT-7700 TEM at 80 kV.

For WT and Tud domain mutants, size (area) of multiple individual polar granules was measured from different electron micrographs using ImageJ 1.53t (NIH), similarly to previous characterization of earlier set of X-ray generated *tud* alleles (Boswell & Mahowald, 1985). Unpaired two-tailed *t* test was used to determine if the differences in the size of the granules in the mutants are statistically significant from that in the WT control (Fig 5E).

# Data Availability

All data that support the conclusions of this work are available within the article, and any additional relevant information, including the source data, can be obtained from the corresponding author upon reasonable request.

# Supplementary Information

# Acknowledgements

We thank Wahiduzzaman for the production of Tud domains 7–11 fragment used for the generation of anti-Tud antibody and E Hackney for his comments on the manuscript. This study was supported by a grant from National Science Foundation MCB-2130162 to AL Arkov. Also, this work was supported in part by a grant from the NIH National Institute of General Medical Sciences, P20GM103436.

## Author Contributions

SJ Tindell: formal analysis, investigation, methodology, and writing—review and editing.
AG Boeving: formal analysis, investigation, methodology, and writing—review and editing.
J Aebersold: formal analysis, investigation, methodology, and writing—review and editing.
AL Arkov: conceptualization, formal analysis, supervision, funding acquisition, investigation, methodology, and writing—original draft, review, and editing.

## Conflict of Interest Statement

The authors declare that they have no conflict of interest.

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
