## [Reviewer comments · Life Science Alliance]

Life Science Alliance

Multiple domains of scaffold Tudor protein play non-redundant roles in *Drosophila* germline

Samuel Tindell, Alyssa Boeving, Julia Aebersold, and Alexey Arkov

DOI: <https://doi.org/10.26508/lsa.202503304>

Corresponding author(s): Alexey Arkov, Murray State University

Review Timeline:

Submission Date:	2025-03-12
Editorial Decision:	2025-04-28
Revision Received:	2025-05-21
Editorial Decision:	2025-06-25
Revision Received:	2025-06-30
Accepted:	2025-07-02

Scientific Editor: Sarita Hebbar

Transaction Report:

April 28, 2025

Re: Life Science Alliance manuscript #LSA-2025-03304

Prof. Alexey L. Arkov
Murray State University
Biological Sciences
2112 Biology Building
Murray, Kentucky 42071

Dear Dr. Arkov,

Thank you for submitting your manuscript entitled "Multiple domains of scaffold Tudor protein play non-redundant roles in *Drosophila* germline" to Life Science Alliance (LSA). The manuscript was assessed by three expert reviewers, whose comments are appended to this letter.

All three reviewers commented on the value of this work to the community. That said, we agree with the reviewers that the manuscript needs to be revised with experiments and textual edits before publication at LSA. A revised manuscript must include:

1. Application of phenotypic analyses (in Figures 2-5) on previously described transgenes (Reviewer 3, point 4).
2. Phenotype upon reduction in wild-type Tudor protein levels (Reviewer 2, paragraph 2).
3. Possible role on the valence of the Tudor domains either shown experimentally (Reviewer 3, point 1) or elaborated further in the discussion.
4. Imaging data and quantification related to Figure 3 and 5 (Reviewer 1, point 2, Reviewer 2, points 5-6, Reviewer 3, points 6, 9)
5. Details of the mutagenesis and accurate representation of the location of mutations (Reviewer 2, point 1 and Reviewer 3, point 3)

In line with the overall recommendations, we invite you to submit a revised manuscript addressing the Reviewers' comments. When submitting the revision, please include a letter addressing the reviewers' comments point by point. While a rebuttal must respond to all points in some form, additional data to resolve these points (other than ones indicated above) are not required.

We hope that the comments below will prove constructive as your work progresses. Thank you for this interesting contribution to Life Science Alliance. We are looking forward to receiving your revised manuscript.

Sincerely,

Sarita Hebbar, PhD
Scientific Editor
Life Science Alliance
<http://www.lsajournal.org>

- A letter addressing the reviewers' comments point by point.
- An editable version of the final text (.DOC or .DOCX) is needed for copyediting (no PDFs).
- High-resolution figure, supplementary figure and video files uploaded as individual files: See our detailed guidelines for

preparing your production-ready images, <https://www.life-science-alliance.org/authors>

B. MANUSCRIPT ORGANIZATION AND FORMATTING:

Reviewer #1 (Comments to the Authors (Required)):

Using a CRISPR/Cas9 deletion approach, the authors investigate the role of Tudor domains 2-5 in germ cell formation. They find that each of these Tudor domains contributes to achieving a wild-type number of germ cells, indicating that all domains cooperate for optimal germ cell formation. Surprisingly, however, the deletion of individual domains results in phenotypes of varying severity, affecting both the accumulation of germ plasm components and the number of germ cells formed. These results provide new insights into how the 11-domain protein scaffold, Tudor, organizes germ plasm to ensure optimal germ cell formation.

Overall, I only have two smaller comments:

- 1) Figure 2: Why were germ cells counted at nuclear cycles 10-13 instead of cycle 14, when germ cells stop dividing and reach their final number? While more challenging, counting at cycle 14 is feasible and would provide a more definitive assessment of germ cell numbers. Additionally, since it is unclear whether the CRISPR/Cas9-induced mutations affect other developmental processes beyond germ cell formation, it cannot be ruled out that these edits may have stalled embryonic development at an earlier nuclear cycle, leading to an artificially reduced germ cell count simply because division had not yet concluded. Furthermore, have the authors back-crossed the flies to eliminate potential off-target effects of CRISPR/Cas9?
- 2) Figure 5 is missing a control staining without the primary antibody and only with the secondary labeled to show non-specific staining.

Reviewer #2 (Comments to the Authors (Required)):

Comments

This manuscript investigates the Tudor (Tud) protein, which contains 11 Tudor domains and is essential for germ granule formation. The authors generated mutants lacking Tudor domains 2, 3, 4, and 5 individually and analyzed their phenotypes in embryos. The mutants showed reduced or complete loss of germ cells, and some exhibited altered localization of Tud to germ granules. Western blot analysis confirmed a significant reduction in Tud protein levels in the ovaries of these mutants. Electron microscopy revealed that mutants of Tudor domains 2, 4, and 5 had smaller germ granules in embryos. Combined with previous studies, this work suggests that each of the 11 Tudor domains has a unique function contributing to germ cell formation.

The manuscript presents interesting findings on the domain-specific functions of Tudor, but a serious concern remains: the authors cannot exclude the possibility that the reduction in Tud protein levels, rather than the loss of domain function, perturbed germ plasm formation. The reviewer strongly recommends addressing this issue by expressing a reduced dosage of wild-type Tudor (only one copy over a Df) or by using a weak promoter. The authors should examine pole plasm formation with lower levels of wild-type Tud protein. Addressing this issue as well as those below would significantly strengthen the conclusions and improve the clarity of the work.

Major Comments

1. Figure S1A shows the mutation positions, but the exact amino acid boundaries of each Tudor domain (1-11) should be clearly specified. Without this information, the reason for the determination of specific deletion sites in Tudor domains 2-5 is unclear. In addition, Tudor domain can be further categorized into two different groups: canonical Tudor domains and extended Tudor domains, which contain an additional α -helix and two β -strands at N-terminal to the Tudor core and several helices and strands at its C-terminus. The authors should introduce these criteria and categorize the domains accordingly in the main text or supplementary materials.
2. Figure 1B shows the methylarginine binding pocket in Tud11, but there is minimal discussion about its conservation in Tud2-5. The authors should elaborate on whether the lack of conservation influenced their decision to use complete deletions rather than point mutations.
3. To substantiate the claim about reduced protein stability (Lines 163-166), the authors should demonstrate that mRNA expression levels remain unchanged.
4. There is a wrong statement in Lines 151-152 (claiming "Tud does not localize to germ plasm"). This conflict should be amended.
5. In Figure 3, while the authors claim that localization of Tud2, 3, and 5 to germ plasm is lost, Tud staining (in red) in panels B, C, and D is discernible around the germ plasm area that may be background membrane staining. Figure S2 clearly shows the antibody specificity and no background. The authors should provide any better images for Figure 3. Alternatively, co-staining with a membrane marker would clarify. Furthermore, the Vas signal in panels B and C appears more diffuse than in panel A, which may need explanation.
6. For Figure 5, the claim that polar granules are smaller in mutants would be more convincing with statistical analysis of granule sizes.

Minor Comments

1. The manuscript uses inconsistent terminology for developmental stages across figures. Figure 2A-E and Figure 3 refer to "early blastoderm embryos," Figure 2F mentions "Stages 10-13" and "stages 5-14," while Figure 5 uses "preblastoderm embryos." These should be standardized or properly corrected.
2. Lines 112-113 should be rephrased to avoid the misinterpretation that a single mutant contains all mutations.
3. In Figure 2F, the authors should include the count numbers for tudom3 mutants. In addition, the germ cell counting methodology mentioned in the figure legend should be detailed in the Materials and Methods section.
4. In Figure 4, the wild-type control phenotype should be clearly specified (homozygous or wt/Df).
5. In Lines 288-291, the basis for the statement that granules "appear to contain mutant Tud protein" is not clear and requires further explanation or evidence.

Reviewer #3 (Comments to the Authors (Required)):

This manuscript concerns the *Drosophila* Tudor (Tud) protein and aims to address the role of specific Tudor domains. Tud is a major scaffold protein for the formation of germ granules in *Drosophila* oocytes and embryos. Germ granules are membraneless condensates and serve as models to understand the formation, organization and function of membraneless condensates. *Drosophila* Tud contains 11 Tudor domains, several of which have been analyzed through various approaches, i.e. utilization of mutants or transgenic lines and structural analysis. However, Tudor domains 2 to 6 have not been analyzed previously. Using the CRISPR approach, the authors produce deletions of each of these Tudor domains and address their function and potential redundancy *in vivo*. Given the importance of Tud and Tudor domain-containing proteins in condensate formation, the question is interesting and might provide information of broad interest for biomolecular condensate biology. The data and writing of the manuscript should be greatly improved. Additional experiments are required to firmly conclude about redundancy. These concerns, as detailed below, should be addressed before publication.

Major concerns.

- 1) The main concern is that the conclusion, namely that the different Tudor domains do not act redundantly, is actually not established from the presented data. Deletion of Tudor domains 2, 4 and 5 produces largely similar phenotypes. The authors conclude that each Tudor domain has a specific function and is not redundant with the other domains. However, another possibility would be that for a wild-type Tud function in germ granule organization, Tud valence is essential. In other words, having 11 Tudor domains with 11 possibilities of protein interaction would be compulsory to produce germ granule with a wild-type structure (see for example Sanders DW et al. Cell 2020). This question should be addressed by replacing one of the Tudor domains by another one (among domains 2, 4, 5), maintaining the valence while removing a specific domain. This question is important since the valence of scaffold proteins is key to condensation.
- 2) The Introduction should be improved to at least introduce germ granule composition; the basis of Tud/Aub interaction; the relationships between germ granule composition/function and the development of primordial germ cells; all those being major aspects of the manuscript.
- 3) Lack of domain 6 deletion: It would be useful that the authors provide an explanation. The mutagenesis design is not detailed. Is it possible that a deletion of domain 6 was not retrieved because mutant individuals die and could not be kept through the

mutagenesis crosses? Would this indicate a dominant lethal mutation since tud1 is not lethal? Any other explanation?

4) Because the phenotypic analyses are not exactly the same in different publications, it would have been useful to run the different experiments in parallel with previously studied transgenes such as mini-tud 3 (Arkov et al. Development 2006) and tud7-11 (Liu et al. Genes Dev. 2010) that seem to produce different phenotypes although they express almost the same Tudor domains.

5) It would be useful to summarize what is known of the mechanisms leading to Tud localization to the germ plasm. Also, using the same term "localization" throughout the manuscript, instead of "transport" would improve clarity, unless something is known about Tud transport.

6) Quantification of data in Fig. 3 is required, both for the level of Tud localized in the germ plasm and the number of embryos showing the phenotype. Also, could the lack of Tud enrichment in the germ plasm be analyzed using Airyscan microscopy in tud[dom2] and tud[dom5] mutants. If there is no enrichment of Tud in the germ plasm, where is Tud? We would expect a very low level of Tud in the germ plasm if indeed Tud is diffuse in the whole embryo, but this is not what we see in Fig. 5.

7) A table recapitulating the different phenotypes identified (from Fig. 2 to Fig. 5) for all four mutants would be useful.

8) p. 9: The type of microscopy (Airyscan) used should be indicated in the text for clarification.

9) To which level the different Tud mutant proteins are incorporated in germ granules? In Fig. 5 Vas-Tud colocalization seems to be lower in tud mutants than in wild-type embryos. Colocalization should be quantified. Germ granules could be better defined using colocalization of Tud or FLAG with Aub and Osk.

10) Germ granules visualized using EM appear to be more affected in tud[dom4] than in other mutants, or at least different (possibly hollow?). Could this be confirmed?

11) Table 1. Tudor domain 1: small germ granules (Ramat et al. Nature Comm 2024)

Is there evidence that Tudor domain 2 does not bind Aub?

Tudor domain 3: no germ cell; no Tud protein detected

12) Discussion: More than the size of germ granules, defects in their function would affect germ cell development and therefore the number of germ cells. Defects in translational activation have been shown in tudA36 (Ramat et al. Nature Comm 2024). In general a lack of reflection and/or information throughout the manuscript decreases its interest and impact.

13) Could structural information regarding different Tudor domains and residues described in (Liu et al. Genes Dev 2010) and (Ren et al. Cell Res 2014) be used to further interpret data obtained in this manuscript.

Minor comments.

1) To increase the uniformity between germ granules in different species, and improve all readers' understanding, it would be better to use the term "germ granules", not polar granules for Drosophila as well.

2) It would be more logical to name these mutants tud[dom2](Delta-dom2), tud[dom3]...

Dear Dr. Hebbbar,

We appreciate your work on our manuscript. Also, we would like to thank the reviewers for their work and comments. We have now carefully considered and addressed all points raised by reviewers and according to your suggestions, we have revised our manuscript. In particular, we have included additional data on quantification of microscopy images, described details of mutagenesis and Tud domains characterized in our work and provided additional analysis and discussion in the context of previous research. Our revision resulted in a new figure panel (Fig. 5E) and two new supplementary tables (Table S1 and S2) as well as additions and updates in the text. Our specific changes are described below and our responses to the comments of the reviewers are divided in two groups. First, we will address the comments of the reviewers that you highlighted followed by our responses to all additional comments from the reviewers.

Editor

'1. Application of phenotypic analyses (in Figures 2-5) on previously described transgenes (Reviewer 3, point 4).'

Reviewer 3, point 4

'4) Because the phenotypic analyses are not exactly the same in different publications, it would have been useful to run the different experiments in parallel with previously studied transgenes such as mini-tud Δ 3 (Arkov et al. Development 2006) and tud7-11 (Liu et al. Genes Dev. 2010) that seem to produce different phenotypes although they express almost the same Tudor domains.'

While we have done quantitative phenotypic analysis of *mini-tud Δ 3* transgene, which shows a reduced capacity of this transgene to form germ cells (Arkov et al. Development 2006), *tud7-11* (Liu et al. Genes Dev. 2010) transgene has not been similarly quantified and other than both of these two transgenes can rescue some germ cell formation in *tud* mutant background when expressed from non-native genomic loci, no quantitative differences between these transgenes can be concluded. Also, in addition to domains 7-11, exclusively expressed by *tud7-11* transgene, *mini-tud Δ 3* transgene expresses Tud domain 1 and parts of the linkers between Tud domains 1 and 2 and Tud domains 6 and 7. Therefore, if there are indeed some potential differences in the phenotypes, these differences might be caused by these regions that are missing in *tud7-11* transgene or by potentially different expression levels of these transgenes. Also, data on *tud7-11* transgene were published by other researchers and we do not have access to that line in Arkov lab. However, given that there could be multiple reasons for causing potential differences between different transgenes and the focus of our current work on characterization of new *tud* alleles expressed from native *tud* locus, we do not think that these additional experiments on the comparison of transgenes would lead to conclusive data and are needed for our current manuscript.

Editor

'2. Phenotype upon reduction in wild-type Tudor protein levels (Reviewer 2, paragraph 2).'

Reviewer 2, paragraph 2

'The manuscript presents interesting findings on the domain-specific functions of Tudor, but a serious concern remains: the authors cannot exclude the possibility that the reduction in Tud protein levels, rather than the loss of domain function, perturbed germ plasm formation. The reviewer strongly recommends addressing this issue by expressing a reduced dosage of wild-type Tudor (only one copy over a Df) or by using a weak promoter. The authors should examine pole plasm formation with lower levels of wild-type Tud protein.'

The suggested experiment to reduce *tud* dosage by generating flies with one copy of *tud* gene over RNA-null *tud* mutant has been done and the results were published (Thomson, T., Liu, N., Arkov, A., Lehmann, R. & Lasko, P. (2008) Isolation of new polar granule components in *Drosophila* reveals P body and ER associated proteins, *Mech Dev.* **125**, 865-73). These experiments revealed virtually no reduction in germ cell number by reducing the gene copy number indicating that *tud* function in germ plasm assembly and germ cell formation phenotypes are not very sensitive to reduction in its gene dosage. Furthermore, comparable (~50%) reduction of mutant Tud levels in different *tud* mutants cause different effects: Tud enrichment defects in germ plasm for Tud domain 2, 5 mutants with less defective germ cell formation compared with germ plasm enrichment but much stronger defects in germ cell formation for Tud domain 4 mutant (Figs. 2-4), arguing for specific different functions of certain Tud domains not caused merely by reduction in protein levels. Furthermore, if lack of one out of 11 Tud domains causes dramatic decrease in protein levels (for example, in Tud domain 3 mutant (Fig. 4C)), this is an interesting and surprising result in itself, which suggests a critical function of this single 57 amino acid canonical Tud domain in expression/stability of 2515 amino acid Tud protein, which could be defective in this mutant due to several reasons. These reasons may include inability of this mutant Tud to associate with Tud domain 3-specific interacting protein needed to stabilize Tud, critical function of the domain in the assembly of Tud to germ granules or in demixing of Tud molecules from the cytosol (where it might be degraded) into the granules. These points are now discussed in the revision (see Discussion, lines 218-247).

Editor

'3. Possible role on the valence of the Tudor domains either shown experimentally (Reviewer 3, point 1) or elaborated further in the discussion.'

Reviewer 3, point 1

'1) The main concern is that the conclusion, namely that the different Tudor domains do not act redundantly, is actually not established from the presented data. Deletion of Tudor domains 2, 4 and 5 produces largely similar phenotypes. The authors conclude that each Tudor domain has a specific function and is not redundant with the other domains. However, another possibility would be that for a wild-type Tud function in germ

granule organization, Tud valence is essential. In other words, having 11 Tudor domains with 11 possibilities of protein interaction would be compulsory to produce germ granule with a wild-type structure (see for example Sanders DW et al. Cell 2020). This question should be addressed by replacing one of the Tudor domains by another one (among domains 2, 4, 5), maintaining the valence while removing a specific domain. This question is important since the valence of scaffold proteins is key to condensation.'

We appreciate this comment of the reviewer on Tud valence and have elaborated on that idea and referred to the work of Sanders et al., 2020 in Discussion as suggested by the editor (lines 194-202). There are three points that we would like to highlight in response to the comment on valence. First, while Tud scaffold may require a certain number of Tud domains to ensure sufficient valence, in our current work we analyze effect of mutations in single domains that would reduce the valence from 11 to 10 since in each mutant, there are still 10 Tud domains remaining. Therefore, this ~10% reduction in Tud valence in Tud domain mutants by itself cannot explain much larger effects of the mutations that we see in vivo. Second, as described in our response to previous comment, mutations in certain domains, do have different phenotypic effects - ~two-fold reduction in the number of primordial germ cells and defects in Tud germ plasm enrichment in Tud domain 2 and 5 mutants versus ~5-fold reduction in the number of germ cells in just 12% of all embryos (the rest of the embryos failed to form germ cells) and strong Tud enrichment in the germ plasm in Tud domain 4 mutant (Figs. 2 and 3). Finally, we revised the manuscript to clarify that the term 'non-redundancy' is used in our work to mean that any given single Tud domain contributes to the normal function and expression of Tud protein since mutations in any single domain result in significant defects during germ cell development (lines 190-193).

Editor

'4. Imaging data and quantification related to Figure 3 and 5 (Reviewer 1, point 2, Reviewer 2, points 5-6, Reviewer 3, points 6, 9)'

Reviewer 1, point 2

'2) Figure 5 is missing a control staining without the primary antibody and only with the secondary labeled to show non-specific staining.'

Fig. 5 shows posterior germ plasm immunostainings using anti-Vas and anti-FLAG (for FLAG-tagged Tud) antibodies and the specificity of these antibodies for these proteins localized to the germ plasm and germ granules, has been validated in previous publications (for example, more recently for anti-Vas – 1) Trcek T, Grosch M, York A, Shroff H, Lionnet T, Lehmann R (2015) Drosophila germ granules are structured and contain homotypic mRNA clusters. *Nat Commun* 6: 7962. doi:10.1038/ncomms8962; 2) Zheng J, Gao M, Huynh N, Tindell SJ, Vo HDL, McDonald WH, Arkov AL (2016) In vivo mapping of a dynamic ribonucleoprotein granule interactome in early Drosophila embryos. *FEBS Open Bio*: doi: 10.1002/2211-5463.12144. doi:10.1002/2211-5463.12144; and for FLAG-tagged Tud - Wahiduzzaman, Tindell SJ, Alexander E, Hackney E, Kharel K, Schmidtke R, Arkov AL (2024) Drosophila germ granules are

assembled from protein components through different modes of competing interactions with the multi-domain tudor protein. *FEBS Lett*: doi:10.1002/1873-3468.14846). These references are now included in Methods (Immunohistochemistry, lines 308-309).

Also, as evident from this work and as expected from known distribution of Vas, which is the most often used germline marker in *Drosophila* and other animals, the anti-Vas antibody specifically labels germ cells of wild-type embryos (Fig. 2A) and posterior germ plasm of early embryos where Vas is localized before its transport to germ cells (Fig. 3) and this antibody does not label soma in Tud domain 3 mutant embryos (negative control) that fail to form germ cells (Fig. 2C). Similarly, anti-FLAG antibody shows expected specific localization of FLAG-Tud in germ plasm of wild-type embryos (Fig. 3A) and no FLAG-Tud staining of the germ plasm of Tud domain 3 mutant embryos that lack Tud protein (negative control, Fig. 3C, Fig. 4).

Reviewer 2, points 5

'5. In Figure 3, while the authors claim that localization of Tud2, 3, and 5 to germ plasm is lost, Tud staining (in red) in panels B, C, and D is discernible around the germ plasm area that may be background membrane staining. Figure S2 clearly shows the antibody specificity and no background. The authors should provide any better images for Figure 3. Alternatively, co-staining with a membrane marker would clarify. Furthermore, the Vas signal in panels B and C appears more diffuse than in panel A, which may need explanation.'

We agree with the reviewer that the cortical staining is detected in red channel in the Tud domain 2, 3 and 5 mutants in Fig. 3B, C, and E respectively and that this is background staining since it is also detected in *tud* null mutant embryos (Tud domain 3 mutant, Fig. 3C). While we do not know the exact reason for this background staining, we agree that this is a membrane staining since it never colocalizes with Vas in cortical germ plasm (green channel in this figure), which is closely adjacent to the posterior membrane. Images in Fig. S2 mentioned by the reviewer show experiments with different antibody (rabbit anti-Tud antibody). However, in Fig. 3 we are using anti-Vas antibody from rabbit as well and could not use two rabbit antibodies in this particular multi-labeling experiment. However, regardless of the exact origin of this background staining, the images in this figure represent the consistent pattern that we see in the multiple mutant embryos and the germ plasm staining is not affected or obscured by the background staining as we can clearly identify the germ plasm using Vas as a marker. Therefore, we can conclude that, contrary to the wild-type control, in the mutant embryos, there are defects in Tud enrichment in the germ plasm (outlined in the Fig. 3B, C and E).

We do not have evidence that the Vas distribution is different in the mutants compared with the wild-type control and the presented images are likely to show small fluctuations of normal Vas localization to the germ plasm.

Reviewer 2, point 6

'6. For Figure 5, the claim that polar granules are smaller in mutants would be more convincing with statistical analysis of granule sizes.'

Granule size quantification and statistical analysis has now been done (new Fig. 5E) and show that mutant polar granules are significantly smaller than in the wild-type. Fig 5 legend and Methods have been updated to include details on data and methodology.

Reviewer 3, point 6

'6) Quantification of data in Fig. 3 is required, both for the level of Tud localized in the germ plasm and the number of embryos showing the phenotype. Also, could the lack of Tud enrichment in the germ plasm be analyzed using Airyscan microscopy in tud[dom2] and tud[dom5] mutants. If there is no enrichment of Tud in the germ plasm, where is Tud? We would expect a very low level of Tud in the germ plasm if indeed Tud is diffuse in the whole embryo, but this is not what we see in Fig. 5.'

While the fluorescently-labeled Tud in the germ plasm fluctuates and the precise levels of Tud enrichment have not been possible to quantify accurately, Tud is always enriched in the germ plasm of wild-type embryos (described with references in the revision in lines 127-128) and is exclusively required there for the germ cell formation. However, according to the reviewer's comment, we have now quantified the phenotype of Tud localization (lines 534-537 in Fig 3 legend).

Airyscan microscopy images of Tud domain 2 and 5 mutants are shown in Fig 5 and these images indicate that mutant Tud is incorporated into granules. As quantified from our EM data, these granules are very small (new Fig. 5E) and their size is similar to the resolution limit of our Airyscan microscopy (~140 nm in lateral spatial resolution). Therefore, the accurate analysis of the Tud-containing granules' characteristics in the mutants from Airyscan images and how this may relate to lack of Tud enrichment in the germ plasm with Airyscan is very difficult since these granules are diffraction-limited. In other words, although the Tud-containing polar granules in the mutants are very small, they will appear larger in the Airyscan images than they actually are because of diffraction. However, to overcome these light microscopy limitations, we characterized the granules in these mutants with EM and the presence of small but multiple polar granules supports the presence of some mutant Tud protein in the germ plasm which is known to be required for the granule formation. Therefore, our data presented in Figs. 3 and 5, which include fluorescent microscopy and EM results are consistent with each other and suggest that while Tud is not visibly enriched in the germ plasm in Tud domain 2 and 5 mutants (with unlocalized Tud presumably distributed in the rest of the embryo), it is still present there at levels sufficient for its incorporation into small germ granules. These considerations and clarifications have now been included in our revised manuscript (lines 165-169, 176-182).

Reviewer 3, point 9

'9) To which level the different Tud mutant proteins are incorporated in germ granules? In Fig. 5 Vas-Tud colocalization seems to be lower in tud mutants than in wild-type embryos. Colocalization should be quantified. Germ granules could be better defined using colocalization of Tud or FLAG with Aub and Osk.'

As suggested, we quantified the Vas-Tud colocalization and determined the number of Vas granules that also show Tud and, in accordance to the reviewer's suggestion, there appear to be somewhat more Vas granules that do not show Tud in the same granules in the mutants (less colocalization) than in the wild-type (new Table S1). This difference, however, is not statistically significant (P values ranging from 0.17 to 0.34, Table S1). It is possible that this apparent lack of Tud fluorescence in some of Vas granules, could be due to a very low levels of Tud in some granules in the mutants, which, therefore, escape detection.

Editor

'5. Details of the mutagenesis and accurate representation of the location of mutations (Reviewer 2, point 1 and Reviewer 3, point 3)'

Reviewer 2, point 1

'1. Figure S1A shows the mutation positions, but the exact amino acid boundaries of each Tudor domain (1-11) should be clearly specified. Without this information, the reason for the determination of specific deletion sites in Tudor domains 2-5 is unclear. In addition, Tudor domain can be further categorized into two different groups: canonical Tudor domains and extended Tudor domains, which contain an additional α -helix and two β -strands at N-terminal to the Tudor core and several helices and strands at its C-terminus. The authors should introduce these criteria and categorize the domains accordingly in the main text or supplementary materials.'

As suggested, the exact amino acid boundaries of each Tud domain aligned with each other are now specified in Fig. S1B and Fig. 1B. In addition, each of these figures specifies that each of the domain sequences shown and mutated is a canonical Tud domain. The reason for deleting canonical domains is that these small domains contain the principal binding regions for interactions with other proteins. Categories of Tud domains are now discussed in the revision (main text: lines 46-51, 58-62, 79, Fig. 1B legend; supplementary materials; Fig. S1B legend).

Reviewer 3, point 3

'3) Lack of domain 6 deletion: It would be useful that the authors provide an explanation. The mutagenesis design is not detailed. Is it possible that a deletion of domain 6 was not retrieved because mutant individuals die and could not be kept through the mutagenesis crosses? Would this indicate a dominant lethal mutation since tud1 is not lethal? Any other explanation?'

Details of the CRISPR/Cas9 mutagenesis that we used to generate all Tud domain deletions (including unsuccessful mutagenesis of domain 6) are now provided

(revised Methods, in 'Tud domain mutants' section, starting on line 280 and new Table S2). Although for Tud domain 6, we used an additional gRNA (Table S2), we could not succeed with generating deletion of this domain, however, there might not be a biological (genetic) reason for that. Therefore, while we think that the reviewer's questions, regarding this domain are interesting, we are not ready to provide an explanation for the lack of success with this deletion at this moment and more detailed analysis of this domain will have to await further investigation.

This concludes our responses to all the comments highlighted by the editor. Our responses to the additional points mentioned by the reviewers follow.

Reviewer 1

'1) Figure 2: Why were germ cells counted at nuclear cycles 10-13 instead of cycle 14, when germ cells stop dividing and reach their final number? While more challenging, counting at cycle 14 is feasible and would provide a more definitive assessment of germ cell numbers. Additionally, since it is unclear whether the CRISPR/Cas9-induced mutations affect other developmental processes beyond germ cell formation, it cannot be ruled out that these edits may have stalled embryonic development at an earlier nuclear cycle, leading to an artificially reduced germ cell count simply because division had not yet concluded. Furthermore, have the authors back-crossed the flies to eliminate potential off-target effects of CRISPR/Cas9?'

Germ cells were counted at embryonic stages 10-13 when they are spread out and, therefore, can be quantified accurately, not at nuclear cycles 10-13, which occur much earlier during development than the embryonic stages that we used for counting. Therefore, there are no issues, mentioned by the reviewer, which would affect our analysis or conclusions. Also, the flies were back-crossed.

Reviewer 2

Major comments

'2. Figure 1B shows the methylarginine binding pocket in Tud11, but there is minimal discussion about its conservation in Tud2-5. The authors should elaborate on whether the lack of conservation influenced their decision to use complete deletions rather than point mutations.'

The conservation of different Tud domains and their significance for protein-protein interactions are now discussed (lines 493-499). Furthermore, the lack of conservation was not a primary reason for us to delete complete domains. Rather, we aimed at deleting a given domain to determine directly if other 10 remaining domains would be able to maintain Tud function in germ cell formation.

'3. To substantiate the claim about reduced protein stability (Lines 163-166), the authors should demonstrate that mRNA expression levels remain unchanged.'

A change in mRNA expression is not very likely since the introduced small deletion mutations do not affect promoter or other regulatory regions and they do not create premature stop codons, and, therefore, should not result in significant change of RNA levels.

'4. There is a wrong statement in Lines 151-152 (claiming "Tud does not localize to germ plasm"). This conflict should be amended.'

The statement on these lines of the original manuscript refers to the *tud* mutants that do not enrich mutant Tud protein in the germ plasm contrary to the wild-type.

Minor comments

'1. The manuscript uses inconsistent terminology for developmental stages across figures. Figure 2A-E and Figure 3 refer to "early blastoderm embryos," Figure 2F mentions "Stages 10-13" and "stages 5-14," while Figure 5 uses "preblastoderm embryos." These should be standardized or properly correct.'

Done as suggested.

'2. Lines 112-113 should be rephrased to avoid the misinterpretation that a single mutant contains all mutations.'

The sentence is now rephrased (lines 114-115 of revision).

*'3. In Figure 2F, the authors should include the count numbers for *tuddom3* mutants. In addition, the germ cell counting methodology mentioned in the figure legend should be detailed in the Materials and Methods section.'*

All Tud domain 3 mutant embryos failed to make any germ cells. The number of embryos with no germ cells scored for this mutant (n=95) is given in parenthesis below '*tud^{dom3}*' as for other mutants and is specified in the figure legend. 'Materials and Methods' have now include methodology for germ cell counting (lines 310-314).

'4. In Figure 4, the wild-type control phenotype should be clearly specified (homozygous or wt/Df).'

Done (line 541).

'5. In Lines 288-291, the basis for the statement that granules "appear to contain mutant Tud protein" is not clear and requires further explanation or evidence.'

According to this comment, this statement has been clarified (lines 253-255).

Reviewer 3

Major comments

'2) The Introduction should be improved to at least introduce germ granule composition; the basis of Tud/Aub interaction; the relationships between germ granule composition/function and the development of primordial germ cells; all those being major aspects of the manuscript.'

The introduction has now been revised to include some more general aspects suggested by the reviewer and additional references but also to keep focus on Tud scaffold and Tud domains, which are at the core of the present study.

'5) It would be useful to summarize what is known of the mechanisms leading to Tud localization to the germ plasm. Also, using the same term "localization" throughout the manuscript, instead of "transport" would improve clarity, unless something is known about Tud transport.'

Unfortunately, the mechanism leading to Tud localization to germ plasm is not understood and this will be an interesting subject of future research. Also, the suggested change of the term "transport" to "localization" has been done in the revision.

'7) A table recapitulating the different phenotypes identified (from Fig. 2 to Fig. 5) for all four mutants would be useful.'

Since we include Table 1 in the main text, which lists essential aspects of all known Tud domain mutants (including the ones characterized in this work), we believe that an additional table is not necessary.

'8) p. 9: The type of microscopy (Airyscan) used should be indicated in the text for clarification.'

Done as suggested (line 157).

'10) Germ granules visualized using EM appear to be more affected in tud[dom4] than in other mutants, or at least different (possibly hollow?). Could this be confirmed?'

We have quantified the size of the granules in the revision (new Fig. 5E) and the granules in all the mutants (including Tud dom 4 mutant) have very similar size. Also, granule morphology in all these mutants looks similar.

'11) Table 1. Tudor domain 1: small germ granules (Ramat et al. Nature Comm 2024) Is there evidence that Tudor domain 2 does not bind Aub? Tudor domain 3: no germ cell; no Tud protein detected.'

Tud domain 1 entry is now revised as suggested. Indeed, Tud domain 2 failed to bind to Aub in in vitro binding assay (Vo HDL, Wahiduzzaman, Tindell SJ, Zheng J, Gao M, Arkov AL (2019) Protein components of ribonucleoprotein granules from *Drosophila* germ cells oligomerize and show distinct spatial organization during germline development. *Sci Rep* 9: 19190. doi:10.1038/s41598-019-55747-x). Also, given that for many previously characterized mutants, there is no quantification of the germ cell formation phenotype, we do not attempt to detail the strength of this phenotype for all the mutants in the Table 1 as specified in a caption below the table. The lack of detectable Tud domain 3 mutant protein is now indicated in the Table 1 as suggested.

'12) Discussion: More than the size of germ granules, defects in their function would affect germ cell development and therefore the number of germ cells. Defects in translational activation have been shown in tudA36 (Ramat et al. Nature Comm 2024). In general a lack of reflection and/or information throughout the manuscript decreases its interest and impact.'

Discussion has been expanded to include different aspects raised by this reviewer and reviewer 2. Also, functional implications of the granules assembled in mutants and data on defective translational activation observed in *tud^{A36}* mutant are now discussed in the context of data presented in this work as suggested.

'13) Could structural information regarding different Tudor domains and residues described in (Liu et al. Genes Dev 2010) and (Ren et al. Cell Res 2014) be used to further interpret data obtained in this manuscript.'

Structural and biochemical analysis of Tud domains characterized in this work and their binding partners would ultimately be needed to provide molecular insights into mutant phenotypes observed in our work. However, we have included a more specific analysis of different Tud domains and their potential binding properties in the context of known structural information of Tud domains 9-11 reported previously in publications mentioned by the reviewer (lines 493-499).

Minor comments

*'1) To increase the uniformity between germ granules in different species, and improve all readers' understanding, it would be better to use the term "germ granules", not polar granules for *Drosophila* as well.'*

In *Drosophila*, there are two major types of germ granules: polar granules and 'nuage' – perinuclear structure formed around nurse cells' nuclei in developing egg chambers in the ovary. Therefore, to distinguish between these two types and avoid any confusion, we would prefer to keep using this classic term 'polar granules'.

'2) It would be more logical to name these mutants tud[Δdom2](Delta-dom2), tud[Δdom3]'

Since 'delta Δ' has already been taken to name various *tud* transgenes (for example, *mini-tud*Δ1, Δ2 and Δ3), we have avoided using 'Δ' in the names of our Tud domain mutants described in this work.

On behalf of all authors, I would like to thank you and the reviewers again for the comments and work on our manuscript.

Sincerely,

Alexey Arkov

June 24, 2025

RE: Life Science Alliance Manuscript #LSA-2025-03304R

Prof. Alexey L. Arkov
Murray State University
Biological Sciences
2112 Biology Building
Murray, Kentucky 42071

Dear Dr. Arkov,

Thank you for submitting your revised manuscript entitled "Multiple domains of scaffold Tudor protein play non-redundant roles in *Drosophila* germline". We would be happy to publish your paper in Life Science Alliance pending final revisions necessary to meet our formatting guidelines.

- Please specify the nature of the wt/control fly strain used in this study. Although wt/control is defined in some figure legends, it is unclear for the legend as described for Figures 4, 5.
- Please add callouts for Figures 1A-B; 2A-B,D-E; S1A-B and S2A-B to your main manuscript text
- Please include a 'Data Availability' section. Please also specify if you are willing to provide source data or have done so in supplementary information.
- Please upload your Supplementary Tables in editable .doc or Excel format.
- Please add the X and Bluesky handles of your host institute/organization, as well as your own and/or one of the authors, in our system
- Please be sure that the authorship listing and order is correct

A. FINAL FILES:

B. MANUSCRIPT ORGANIZATION AND FORMATTING:

spreadsheets for the main figures of the manuscript. If you would like to add source data, we would welcome one PDF/Excel-file per figure for this information. These files will be linked as supplementary "Source Data" files.

Sincerely,
Sarita

Sarita Hebbar, PhD
Scientific Editor
Life Science Alliance
<http://www.lsajournal.org>

Reviewer #2 (Comments to the Authors (Required)):

The authors have adequately addressed all major concerns raised in the initial review. Their clarifications, additional analyses, and revised discussion substantially strengthen the manuscript.

Reviewer #3 (Comments to the Authors (Required)):

The authors have addressed all my concerns. I have no further comments.

July 1, 2025

RE: Life Science Alliance Manuscript #LSA-2025-03304RR

Prof. Alexey L. Arkov
Murray State University
Biological Sciences
2112 Biology Building
Murray, Kentucky 42071

Dear Dr. Arkov,

Thank you for submitting your Research Article entitled "Multiple domains of scaffold Tudor protein play non-redundant roles in *Drosophila* germline". It is a pleasure to let you know that your manuscript is now accepted for publication in Life Science Alliance. Congratulations on this interesting work.

DISTRIBUTION OF MATERIALS:

I hope you found the review process to be constructive and are pleased with how the manuscript was handled editorially. We look forward to future exciting submissions from your lab.

Sincerely,
Sarita

Sarita Hebbar, PhD
Scientific Editor
Life Science Alliance
<http://www.lsajournal.org>